# Study on Linewidth and Phase Noise Characteristics of a Narrow Linewidth External Cavity Diode Laser

**DOI:** 10.3390/s24041103

**Published:** 2024-02-08

**Authors:** Sheng Hu, Puchu Lv, Chenggang Guan, Shasha Li, Haixin Qin, Xiaoqiang Li, Xuan Chen, Linfeng Zhan, Weiqi Wang, Yifan Xiao, Minghu Wu

**Affiliations:** 1Laboratory of Optoelectronics and Sensor (OES Lab), School of Science, Hubei University of Technology, Wuhan 430068, China; hus@oeslab.com.cn (S.H.);; 2School of Electrical and Electronics Engineering, Hubei University of Technology, Wuhan 430068, China; 3AOV Energy LLC, Wuhan 430068, China

**Keywords:** inter-satellite laser communication, external cavity diode laser, linewidth, phase noise, F-P etalon, optimal operating point

## Abstract

In the field of inter-satellite laser communication, achieving high-quality communication and compensating for the Doppler frequency shift caused by relative motion necessitate lasers with narrow linewidths, low phase noise, and the ability to achieve mode-hop-free tuning within a specific range. To this end, this paper investigates a novel external cavity diode laser (ECDL) with a frequency-selective F-P etalon structure, leveraging the external cavity F-P etalon structure in conjunction with an auxiliary filter to achieve single longitudinal mode selection. The laser undergoes linewidth testing using a delayed self-heterodyne beating method, followed by the testing of its phase noise and frequency noise characteristics using a noise analyzer, yielding beat spectra and noise power spectral density profiles. Furthermore, the paper introduces an innovative bidirectional temperature-scanning laser method to achieve optimal laser-operating point selection and mode-hop-free tuning. The experimental results showcase that the single longitudinal mode spectral side-mode suppression ratio (SMSR) is around 70 dB, and the output power exceeds 10 mW. Enhancing the precision of the F-P etalon leads to a more pronounced suppression of low-frequency phase noise, reducing the Lorentzian linewidth from the initial 10 kHz level to a remarkable 5 kHz level. The bidirectional temperature-scanning laser method not only allows for the selection of the optimal operating point but also enables mode-hop-free tuning within 160 pm.

## 1. Introduction

The laser is the core component in inter-satellite laser communication systems, primarily employed for the generation and receiving of optical signals. Inter-satellite laser communication imposes stringent requirements on system performance metrics such as data capacity, phase noise, and sensitivity. Moreover, due to relative motion, Doppler frequency shift may occur, necessitating laser attributes including narrow linewidth, low phase noise, high frequency stability, and tunability [1,2,3]. Narrow linewidth semiconductor lasers, with advantages including compact size, extended operational lifespan, ease of integration, and direct modulation capability, have emerged as ideal light source devices in fields like inter-satellite laser communication [4,5,6].

In satellite laser communication systems, the choice of communication wavelengths has primarily focused on 1064 nm and 1550 nm in recent years. This is intended to achieve larger antenna gains, minimize signal pointing attenuation, and reduce the impact of solar background and sunlight scattering. As technology advances, compatibility with both 1064 nm and 1550 nm communication wavelengths is gradually becoming the development trend for future satellite laser communication systems [7]. Considering the wider application of the 1550 nm laser communication wavelength and the mature preparation of industrial-grade optical components at this wavelength, this paper selects a narrow linewidth external-cavity laser operating in the 1550 nm band for investigation.

Existing research indicates that the bit error rate (BER) of space coherent optical communication systems is highly sensitive to changes in laser linewidth. As the laser linewidth increases, the phase noise also increases, leading to a corresponding decrease in the system’s signal-to-noise ratio (SNR) and an increase in the BER. Moreover, in cases where the laser beam experiences significant jitter and boresight errors, a laser with a linewidth less than 8 kHz is required to meet the BER requirements for high-quality communication [8]. Additionally, Li, L. et al. investigated an 8-QAM coherent free-space optical communication system employing amplitude compensation and phase recovery, where the combined linewidth tolerance of the transmitter and local oscillator is 10 kHz [9]. Therefore, in terms of the linewidth of lasers, it is essential to study lasers with a linewidth less than 10 kHz to better meet the requirements of inter-satellite laser communication in terms of communication distance and quality.

The concept of linewidth was first introduced in 1958 by the inventors of lasers, Schawlow and Townes, along with the classical Schawlow–Townes linewidth calculation formula [10]. However, subsequent experimental studies revealed a significant discrepancy between the theoretically calculated linewidth based on the Schawlow–Townes formula and the actual measurements of lasers. In 1982, C.H. Henry and colleagues proposed a modification to the theory of semiconductor laser linewidth, emphasizing that adjusting the laser cavity length through appropriate means, such as introducing optical feedback, can narrow the linewidth and enhance output performance [11]. Subsequently, the team led by K.Y. Lious conducted practical measurements and data analysis on lasers with different cavity lengths, leading to the crucial conclusion that linewidth is inversely proportional to output power and cavity length. This laid the foundation for the research and development of external cavity semiconductor lasers and their linewidth and noise suppression [12].

In 2006, the Institute of Semiconductors at the Chinese Academy of Sciences utilized external cavity feedback principles to create an external cavity grating feedback semiconductor laser, reducing the laser linewidth by one order of magnitude and achieving a leap from THz to MHz levels [13]. Since then, various types of external cavity strong feedback structures for narrow linewidth semiconductor lasers have emerged. Through appropriate external cavity feedback methods, the coherent performance of lasers has been significantly improved, leading to the widespread application of semiconductor light sources [14,15,16].

As the research and development technology of narrow linewidth semiconductor lasers has become increasingly mature, their performance has improved significantly, making them a focal point of recent studies. In order to explore their application prospects in the field of inter-satellite laser communication and achieve high-quality communication while compensating for the Doppler frequency shift caused by relative motion, this study investigates a novel low-noise narrow linewidth diode laser with an external cavity structure. By introducing external cavity optical feedback and extending the cavity length, the laser achieves reduced linewidth and noise suppression. This is achieved by utilizing F-P etalons in combination with ultra-narrow bandpass filters for laser frequency selection. Simultaneously, it enables the generation of a high-power single-frequency laser output. Furthermore, based on the standard wavelength table specified by the International Telecommunication Union (ITU), a bidirectional temperature-scanning laser method is employed to select the optimal operating point, thereby achieving wavelength-hop-free tuning within a certain range.

## 2. Basic Principles

### 2.1. Basic Operating Principle of External Cavity Diode Lasers

The external cavity structure refers to the construction of a passive cavity outside the laser diode (LD) using components such as planar mirrors and gratings. Some of the emitted light is reflected into the active region of the LD, creating a composite cavity structure. Diode lasers employing this cavity are collectively known as external cavity diode lasers (ECDLs) [17]. The most fundamental structure, utilizing only planar mirrors to form the external cavity, has been proven in both research and practical applications. Combining an external cavity mirror with the diode laser chip is an effective method for reducing linewidth. The basic model of an ECDL is illustrated in Figure 1 [18,19]. In the figure, $r_{1}$, $r_{2}$, and $r_{3}$ represent the reflectivities of the cavity surfaces for the optical field, l denotes the length of the LD chip (inner cavity length), and d represents the distance from the chip’s right facet to the plane mirror of the external cavity (external cavity length). More complex external cavity structures can be built based on this fundamental model.

This type of ECDL still conforms to the Fabry–Perot (F-P) cavity characteristics, where the laser longitudinal modes formed within the F-P resonant cavity have a fixed correspondence with the cavity length, expressed as Δλ=λ^2/(2nL). Here, Δλ represents the spacing between laser longitudinal modes, L denotes the resonant cavity length, and n is the refractive index inside the cavity. In other words, under a constant refractive index and wavelength conditions, longer cavity lengths result in narrower spacing between longitudinal modes. For a passive F-P external cavity, the longitudinal mode linewidth $\delta_{\lambda }$ satisfies the following relationship [20]:(1)$$\delta_{\lambda }=\frac{\Delta \lambda }{\pi \sqrt{R}/\left(1-R\right)}$$
where R is the cavity surface reflectivity. From this equation, it can be observed that both the cavity surface reflectivity and the cavity length are inversely related to the linewidth of the passive cavity. Therefore, by appropriately increasing the cavity length and cavity surface reflectivity, the linewidth of the corresponding resonator-formed laser longitudinal modes can be reduced.

### 2.2. Principle of F-P Etalon Frequency Selection

The F-P etalon is an optical device that selectively suppresses incident light of different wavelengths using multi-beam interference effects. Its key structure consists of a resonator formed by two parallel reflective surfaces with the same reflectivity. Between the two reflective surfaces, there is an F-P etalon medium with a certain thickness. Typically, the same uniform medium is used for a given etalon, and different materials have different refractive indices. Common choices include air-gap etalons and fused silica solid etalons, among others [21].

Due to the interference effect of light, the F-P etalon has different transmittances for light of different frequencies. Let T represent the transmittance of the F-P etalon for a light beam, and for incident light with a wavelength of λ, it follows [20,22] the equation
(2)$$T(\lambda )=\frac{1}{1+\frac{4R}{{\left(1-R\right)}^{2}}\sin^{2}(\frac{2\pi ndcos\theta 1}{\lambda })}$$
where R represents the reflectivity of the two reflective surfaces of the F-P etalon, d is the thickness of the F-P etalon, and θ1 is the angle of refraction of the incident light, which is commonly approximated as cosθ1 equals 1. It can be observed that the frequency-selective performance of the F-P etalon is primarily determined by these two parameters, R and d. Analysis of this periodic transmittance function reveals that when the wavelength of the incident light beam and the thickness and refractive index of the F-P etalon satisfy the relationship (2πnd)/λ=kπ(k=0,1,2,…), the transmittance reaches its maximum value $T_{max}=1$. Accordingly, within one frequency cycle, the F-P etalon has the maximum transmittance for light at the central frequency. At this point, when the reflectance R is constant, the F-P etalon finesse F, i.e., $4R/{\left(1-R\right)}^{2}$, is fixed. As the wavelength moves away from the central frequency, ${sin}^{2}(2\pi ndcos\theta /\lambda )$ increases. This ultimately results in an increase in the denominator of Equation (2), leading to a decrease in the transmittance of the light wave. The net effect is that the further the wavelength is from the central frequency, the stronger the inhibitory effect of the F-P etalon on it. Therefore, selecting the F-P etalon parameters appropriately in the external cavity of a laser can serve the purpose of longitudinal mode selection.

## 3. Experimental Design and Analysis of Results

### 3.1. Optical Characteristics Analysis of F-P Etalon External Cavity Diode Laser

The complete external cavity optical path structure employed by the F-P etalon external cavity diode laser (referred to as FP-ECDL) is shown in Figure 2, with arrows indicating the direction of the optical path. The gain chip’s left facet, in conjunction with the external cavity mirror, forms the complete laser resonant cavity. The collimation lens, etalon, filter, and mirror together constitute the external cavity structure of the FP-ECDL. The unidirectional isolator for forward transmission prevents backscattered light from re-entering the laser cavity, avoiding interference that could affect the laser’s normal operation. Eventually, the laser output is coupled and focused into an optical fiber through a fiber collimator, with the optical fiber’s end being equipped with a fiber connector as the laser emission port.

The active intracavity portion of the FP-ECDL is composed of a semiconductor gain chip, specifically using Thorlabs ‘s Semiconductor Optical Amplifier (SOA) gain chip. The SOA chip has high reflectivity and anti-reflection coatings on its two ends. One output-coupling facet (corresponding to Figure 2, the right facet of the gain chip) has a reflectance less than 0.01%, meaning nearly no reflection capability. The other reflecting cavity facet (corresponding to Figure 2, the left facet of the gain chip) has a reflectance greater than 90%, and the ridge waveguide on this facet has a width of 5 μm, providing a wide tunable range. Therefore, while the gain chip has a high gain coefficient and can amplify light signals, it does not include a complete resonant cavity and cannot independently generate laser oscillation. However, to effectively achieve strong external feedback and minimize interference from internal cavity longitudinal modes, this type of gain chip is well suited to the basic requirements of FP-ECDL.

The gain chip selected for the experimental preparation of FP-ECDL, under conditions of a 150 mA injection current and 25 °C temperature, yields a complete gain spectrum, as shown in Figure 3, after collimation through a lens and testing with an optical spectrum analyzer. The output power was measured using an optical power meter (selected for the 1550 nm wavelength range), and the recorded power at this moment is −6.043 dBm, approximately 248.713 μW. This is a typical gain spectrum of a semiconductor optical amplifier. Unlike laser emitters with a narrow linewidth, this gain chip exhibits an exceptionally wide gain spectrum covering a wavelength range from 1440 nm to 1600 nm, spanning 160 nm. The spectrometer measures a peak bandwidth of approximately 70 nm, with the center peak wavelength near 1537 nm. The gain spectrum line shape in the peak frequency range is remarkably flat. When the gain spectrum peak has a broader bandwidth, it implies that the gains obtained by all longitudinal modes in that range are generally consistent and evenly distributed. Although there might be intense competition among longitudinal modes, single-mode selection can be achieved through an external cavity frequency-selective optical feedback structure. Ultimately, the overall cavity length of the FP-ECDL is approximately 55 mm. Additionally, considering the potential need for lasers in different wavelength bands for inter-satellite laser communication, adjusting the external cavity frequency-selective structure, such as cavity length, frequency-selective component parameters, etc., can allow the production of FP-ECDLs in different wavelength bands to better meet practical requirements.

Afterward, longitudinal mode selection was performed using a fused silica standard with a free spectral range (FSR) of 100 GHz. The spectrum of the SOA chip is shown in Figure 4. This demonstrates the frequency-selective characteristics of the F-P cavity, where wavelengths corresponding to 1548.860 nm, 1549.660 nm, and 1550.460 nm exhibit minimal loss in longitudinal modes.

After undergoing frequency selection with the etalon, final single longitudinal mode selection is achieved by introducing a filter into the FP-ECDL external cavity. Based on the F-P etalon parameters, a 200 GHz filter matching its free spectral range (FSR) was chosen to suppress longitudinal modes outside the target wavelength with greater loss, ensuring that only the longitudinal mode corresponding to the center wavelength of the filter can lase. Considering both reflected and transmitted output light, the reflectance of the external cavity plane mirror is chosen to be 60%. Under the conditions of an injection current of 150 mA and a chip temperature of 25 °C, the measured lasing spectrum from the spectrometer is illustrated in Figure 5. The central wavelength is 1549.844 nm, and the optical power meter (selecting the 1550 nm wavelength band) measures a laser power of 10.55 mW. The single longitudinal mode characteristics of a single-frequency laser can be measured by the side-mode suppression ratio (SMSR), and a single longitudinal mode can generally be considered when the SMSR is above 50 dB. A higher SMSR indicates better single longitudinal mode characteristics. In Figure 5, the spectrometer measured an SMSR greater than 70 dB for the FP-ECDL laser spectrum operating in continuous-wave lasing mode, specifically, the difference between the blue dashed lines, with an exact value of 77.28 dB. It can be observed that the external cavity frequency-selective structure, incorporating the F-P etalon and filter, exhibits excellent single-mode selection characteristics. Furthermore, there is a nearly identical asymmetric shape near the peak of the laser spectrum (peaks at approximately 12 dB on the left and 20 dB on the right). This is attributed to the influence of various intricate factors on the oscillation mode and performance of the FP-ECDL, such as material characteristics, imperfections in the manufacturing of optical components, thermal effects, etc. These factors contribute to a small segment of optical gain near the peak of the laser spectrum that is not as “smooth”, resulting in these asymmetric spectra. However, due to the extremely subtle nature of this asymmetry, it does not impact the actual performance of the FP-ECDL.

Taking into account the thermal effects and temperature sensitivity of semiconductor lasers, the practical fabrication of FP-ECDL involves the use of a semiconductor thermal electric cooler (TEC) to control the chip temperature. The laser is enclosed with a metal tube along with the external cavity structure. Detailed fabrication processes, such as laser coupling and packaging, are not discussed in this paper. Figure 6 shows an FP-ECDL narrow linewidth single-frequency laser packaged in a butterfly 14-pin metal tube.

By linearly varying the injection current of the FP-ECDL, the emitted optical power was tested at different carrier densities. Figure 7 depicts the FP-ECDL laser power characteristic curve with a changing injection current at 25 °C and a 90% coated reflectance.

### 3.2. Comparison of Line Width and Noise Characteristics of FP-ECDL with Two F-P Etalon Precision Levels

According to the basic principles of ECDL, external cavity feedback has a significant impact on the longitudinal mode characteristics, linewidth, and noise properties of the laser output. For FP-ECDL, the F-P etalon is a crucial component of the external cavity. Therefore, changes in its precision are likely to affect the final laser output performance. The following experimental research focuses on the influence of improved precision in the external cavity F-P etalon on the single longitudinal mode output performance of the laser. As shown in Figure 8, these are the transmittance curves of two coated reflectivity gratings measured by a spectrometer under the same incident light. An increase in reflectivity improves the grating’s precision, sharpens the reflection peak, and enhances its suppression effect on light waves outside the central resonant frequency.

Due to the use of YOKOGAWA’s AQ6370D optical spectrum analyzer with a wavelength resolution of 0.02–0.2 nm in the experiment, it is challenging to distinguish any difference in the single longitudinal mode linewidth formed by the external cavity for two FP etalons based solely on the time-domain spectrum. Beat frequency tests were conducted on the two precision FP-ECDLs using a delayed self-heterodyne linewidth measurement system, as shown in Figure 9. AOM denotes the acousto-optic modulator, and RF is the radiofrequency power supply. Incident light is split into two optical waves by a coupler. One optical wave traverses a delay fiber coil (with a delay time of $\tau_{d}$) and combines with the other optical wave on the opposite side of the coupler, generating a beat frequency signal. This signal is then measured for the linewidth using a photodetector (PD) and a spectrum analyzer (SA). It is important to note that when the laser’s linewidth is around 1 kHz, the necessary length of the delay fiber can extend to hundreds of kilometers. In such cases, considering previously proposed algorithms for curve fitting in short fiber beat frequency measurement methods becomes essential [23]. This allows the obtention of relatively accurate measurement results without requiring long-distance optical fibers.

Here is a brief introduction to its testing principle. Assuming the laser’s output center frequency is fixed, the spectrum of the laser output will broaden due to the presence of phase noise. Let the output optical field of the laser be represented as
(3)$$E_{i}\left(t\right)=\sqrt{P}e^{i(\omega t+\varphi (t))}=E(t)e^{i\omega t}$$
where E(t) and P represent the complex amplitude and optical power of the laser optical field, and ω is the angular frequency of the output light. φ(t) represents the phase variation. Assuming a 50:50 splitting ratio for the fiber coupler that spectrally separates the test laser in the system, the optical fields of the signal light after the AOM and the reference light through the delayed fiber can be represented as follows:(4)$$E_{S}\left(t\right)=\frac{1}{\sqrt{2}}\sqrt{P_{S}}cos(\left(\omega_{S}-\omega_{A}\right)t+\varphi (t))$$
(5)$$E_{D}\left(t\right)=\frac{1}{\sqrt{2}}\sqrt{P_{S}}cos(\left(\omega_{S}(t-\tau_{d}\right)+\varphi (t-\tau_{d}))$$
where $\omega_{A}$ is the frequency shift of the AOM, $\tau_{d}$ is the delay introduced by the delay fiber, and the optical field after the second fiber coupler for the two optical paths is given by
(6)$$E_{B}\left(t\right)=E_{S}\left(t\right)+E_{D}\left(t\right)$$

The photodetector (PD) converts the optical signal into a photocurrent $I\left(t\right)$:(7)$$I\left(t\right)=E_{B}\left(t\right)E_{B}^{*}\left(t\right)$$

The autocorrelation function of the photocurrent is
(8)$$R_{I}\left(t\right)=<E_{B}\left(t\right)E_{B}^{*}\left(t\right)E_{B}\left(t+\tau_{d}\right)E_{B}^{*}\left(t+\tau_{d}\right)>$$

According to the Wiener–Khinchin theorem, the Fourier transform of the autocorrelation function of the photocurrent is the power spectral density function $S_{i}\left(\omega \right)$ [24]:(9)$$S_{i}\left(\omega \right)=\frac{{{\frac{1}{2}P}_{S}^{2}\tau }_{c}}{1+{\left[\tau_{c}\left(\omega_{S}\pm \omega_{A}\right)\right]}^{2}}\{1-[cos\left(\omega_{S}\pm \omega_{A}\right)\tau_{d}+\frac{sin\left(\omega_{S}\pm \omega_{A}\right)\tau_{d}}{\left(\omega_{S}\pm \omega_{A}\right)\tau_{c}}]e^{-\tau_{d}/\tau_{c}}\}+\frac{1}{2}P_{S}^{2}\pi e^{-\tau_{d}/\tau_{c}}\delta \left(\omega_{S}\pm \omega_{A}\right)$$

When the delay of the self-heterodyne $\tau_{d}$ is sufficiently large (the delay fiber is long enough), $\tau_{d}\gg \tau_{c}$, the coherence time of the measured laser and the delay of the self-heterodyne can satisfy $\exp(-\tau_{d}/\tau_{c})\ll 1$. Equation (9) can be simplified to
(10)$$S_{i}\left(\omega \right)=\frac{{\frac{1}{2}P}_{S}^{2}}{1+{\tau_{c}\left(\omega_{S}\pm \omega_{A}\right)}^{2}}$$

In this case, the beat frequency line shape presents a Lorentzian shape, and its 3 dB bandwidth is twice the linewidth of the measured laser [25]. The measurement results of the two FP-ECDLs with different F-P etalon precisions under the same operating conditions are presented in Figure 10. It is evident from the broadening of the beat frequency spectrum lines that the high-F-P-etalon-precision FP-ECDL exhibits a narrower linewidth for a single longitudinal mode laser output.

Next, the lasers with two different F-P etalon precisions were subjected to phase noise and frequency noise characteristic tests under the same operating conditions using a noise test instrument. The basic structure of the noise test instrument is shown in Figure 11.

In Figure 11, FRM stands for the Faraday rotating mirror, PZT is the piezoelectric ceramic used to finely tune the interference arm length, and A/D represents the analog-to-digital converter, while DAQ refers to the data acquisition module. The laser under test, when introduced into the fiber Michelson interferometer with a delay of $\tau_{d}$ through the circulator, generates an interference signal. This signal is converted into an analog electrical signal through photoelectric conversion then transformed into a digital signal by the analog-to-digital converter and inputted into the DAQ. Specialized algorithms are applied to process the digital signal, extracting the phase difference information of the laser under test from the raw data. Subsequently, Fourier transformation is employed to obtain the power spectral density of the phase noise. Since frequency is the derivative of phase, the spectral density function $S_{\varphi }(f)$ of phase noise and the spectral density function $S_{v}\left(f\right)$ of frequency noise are related as follows:(11)$$S_{v}\left(f\right)=f^{2}S_{\varphi }(f)$$

The spectral density function of frequency noise can be obtained by differentiating the phase noise. The frequency noise spectrum of a semiconductor laser can be expressed as follows [25,26]:(12)$$S_{v}\left(f\right){=S}_{0}+\frac{K}{f}+\frac{K^{\prime }}{f^{2}}{=S}_{0}[1+\frac{f_{1}}{f}+{\left(\frac{f_{2}}{f}\right)}^{2}]$$
where $f_{1}=K/S_{0}$, and $f_{2}=\sqrt{K^{\prime }/S_{0}}$. Here, $S_{0}$ represents white noise, and $f_{1}$ and $f_{2}$ are the center frequencies or cutoff frequencies for 1/f noise (flicker noise) and $1/f^{2}$ noise (random-walk noise), respectively. In the high-frequency range where white noise predominantly contributes, the laser linewidth exhibits a Lorentzian distribution, and the linewidth at this point is [26]:(13)$$∆v=\pi S_{0}$$

The phase noise test results for FP-ECDLs with two different F-P etalon precisions are shown in Figure 12. The FP-ECDL with higher F-P etalon precision exhibits superior noise characteristics across the entire frequency range. The overall phase noise is more effectively suppressed, resulting in a lower noise spectrum. Whether in terms of 1/f noise or white noise spectra, the FP-ECDL with higher F-P etalon precision demonstrates lower noise indicators. From the experimental results, it is evident that external cavity feedback significantly influences the noise characteristics of FP-ECDL. By increasing the precision of the etalon, external cavity feedback can be appropriately enhanced, further reducing the phase noise of the output laser.

Next, a comparative analysis of frequency noise and Lorentzian linewidth for lasers with two different F-P etalon precisions is presented. The frequency noise curves in the 10 kHz to 1 MHz frequency range are shown in Figure 13.

Consistent with the analysis of phase noise characteristics, the high-F-P-etalon-precision FP-ECDL also exhibits a lower frequency noise power spectral density. Its white noise spectral density is less than 0.5 kHz, around 360 Hz^2^/Hz. In contrast, the low-precision standard-cavity FP-ECDL corresponds to a white noise spectral density of around 2 kHz. According to Equation (13), the Lorentzian linewidth can be calculated by frequency noise as follows: ${\delta f}_{L}(R=90\%)=1.130\;kHz$ for the high-precision case, which is smaller than ${\delta f}_{L}(R=70\%)=6.283\;kHz$ for the low-precision case. Therefore, the high F-P etalon precision FP-ECDL produces a laser output with a smaller Lorentzian linewidth, indicating a reduction of 5 kHz compared to the low-precision FP-ECDL. This is attributed to the enhanced external cavity feedback achieved by using a higher-coating-reflectivity etalon, effectively increasing the external cavity reflectivity and extending the photon lifetime in the cavity, further narrowing the laser’s longitudinal mode linewidth.

Testing the frequency noise power spectral density of two FP-ECDLs at different operating currents, and estimating the Lorentzian linewidth, and, based on the calculation results, the linewidth characteristics of the two FP-ECDLs under different operating currents, the results are shown in Figure 14. It can be observed that the FP-ECDL with high F-P etalon precision exhibits very small linewidth features. With increased operating current injection, the Lorentzian linewidth can be further reduced to below 0.5 kHz.

The single longitudinal mode laser power output of two FP-ECDLs at different operating currents is depicted in Figure 15. The threshold current for the FP-ECDL is approximately in the range of 40–50 mA. Moreover, with an increase in the precision of the F-P etalon, there is a prolonged effect on the photon lifetime, leading to an enhancement in output power. Additionally, as the power within a certain range is increased, it can narrow the linewidth. Therefore, the results in Figure 15 also partly reflect the variations in the Lorentzian linewidth of FP-ECDLs with different etalon precisions, as shown in Figure 14.

### 3.3. Determination of the Optimal Operating Point for FP-ECDL

Considering the need to select appropriate laser communication bands in the field of inter-satellite laser communication, this paper primarily adopts the ITU-prescribed C, L, and H bands with a 100 GHz frequency spacing [27]. Figure 16 shows some of these channels. Accordingly, the proposed optimal operating point for the FP-ECDL refers to the wavelength closest to the one read at 25 °C among the channel wavelengths when the FP-ECDL operates at a certain temperature and current. At this point, the linewidth is relatively small, and the optical power needs to be greater than 10mW. Considering that the long-term operation of the laser and changes in the external environment can cause slight wavelength fluctuations, a wavelength error less than 1 pm compared to the expected channel wavelength in actual measurements can be considered as meeting the wavelength requirements of the optimal operating point [28].

Choosing the high-F-P-etalon-precision FP-ECDL, the actual temperature scanning curve is shown in Figure 17, with a fixed operating current of 200 mA and a scanning range of 10–35 °C. During both the heating and cooling processes, wavelength jumps can be observed. This is because, with an increase in temperature, the gain peak of the diode laser shifts towards longer wavelengths. Simultaneously, the rise in temperature leads to thermal expansion, causing an increase in the laser cavity length, which also results in a change in the longitudinal mode wavelength. The reverse occurs during the cooling process. Additionally, within a certain range (for the same mode), the longitudinal mode of the diode laser linearly changes with temperature. Beyond this range, the laser will jump to the next mode, causing a sudden wavelength jump. This phenomenon is known as mode hopping. Power is supplied to the FP-ECDL, and its operating temperature is set to 25 °C, with the working point labeled as O. At this point, the wavelength is 1550.5853 nm, and by analysis, it is closest to the H33 channel with a wavelength of 1550.52 nm. Considering O as the initial state, the linear segment passing through O represents the FP-ECDL’s initial mode. Through heating and cooling scans, the wavelength–temperature change rates during heating and cooling are calculated to be 0.195 pm/0.01 °C and 0.198 pm/0.01 °C, respectively.

Since the heating scan precedes the cooling scan, the temperature of FP-ECDL after the scan reaches the minimum value of 10 °C. At this juncture, it is observed that during the heating scan, FP-ECDL undergoes a mode jump to the initial mode at the lowest temperature of approximately 22.3 °C. Setting the temperature to 23 °C and then 25 °C restores the initial mode without the issue of secondary mode hopping caused by overshooting. Returning to the initial O point and setting the temperature to 21.57 °C using the known wavelength-temperature change rate results in a wavelength of 1550.5204 nm, with a wavelength error of only 0.4 pm, meeting the requirement of less than 1 pm. Additionally, using the linewidth testing system mentioned earlier, the Lorentzian linewidth at this working point is approximately 2.5314 kHz, denoted as point A. Testing points P1, P2, and P3, with the same wavelength as A, yields actual linewidths of 3.2019 kHz, 2.9742 kHz, and 2.9713 kHz, respectively, all greater than the linewidth at point A. Since the optical wavelength measurement device used for simultaneous wavelength and power readings may introduce some error at higher power levels, a more precise optical power meter (selecting the 1550 nm wavelength range) is separately employed for power readings at these points. The actual powers are all above 13 mW and show minimal differences. Therefore, in the H33 channel, at an operating current of 200 mA, point A is considered the optimal operating point.

In addition, due to the relative motion between satellites, there is the Doppler effect, resulting in a Doppler frequency shift [29]. Therefore, it is necessary to tune the laser wavelength according to the actual situation. In the context of space coherent laser communication, the tuning range of the laser is generally required to be greater than 50 pm [28]. The FP-ECDL studied in this paper has a tunable range of approximately 160 pm for a single mode. For example, in the above test results, the optimal operating point is at point A. Through heating, it can be tuned to point B with a range of about 94 pm, and through cooling, it can be tuned to point C with a range of about 68 pm.

These results indicate that by employing the bidirectional temperature-scanning method for FP-ECDL, an ideal optimal operating point can be obtained. Furthermore, based on the obtained wavelength–temperature change rate, the precise wavelength tuning of FP-ECDL can be achieved. This suggests significant potential applications for FP-ECDL in the field of inter-satellite laser communication.

## 4. Conclusions

To achieve high-quality inter-satellite laser communication and compensate for the Doppler frequency shift caused by relative motion, this paper investigates a novel Fabry–Perot etalon external cavity diode laser (FP-ECDL). The paper begins by providing an explanation of the specific implementation of the FP-ECDL laser’s external cavity structure and then conducts a performance analysis through optical testing experiments. Specifically, using a delayed self-heterodyne linewidth measurement system and a fiber Michelson interferometer noise testing system, the paper obtains the beat spectra, phase noise, and frequency noise power spectral density curves of the FP-ECDL with two different F-P etalon accuracies. The Lorentzian linewidth of the FP-ECDL laser is calculated from the frequency noise spectrum. Finally, a method involving the bidirectional temperature scanning of the laser is proposed to achieve the selection of the laser’s optimal operating point and wavelength no-mode-hop tuning. The experimental results demonstrate that improving the F-P etalon accuracy (coating reflectivity) enhances the laser’s linewidth and phase noise characteristics, reducing its low-frequency phase noise and Lorentzian linewidth. Moreover, the bidirectional temperature-scanning method for FP-ECDL achieves optimal operating point selection and no-mode-hop tuning within a range of 160 pm.

In the future, achieving more precise, faster, and higher-resolution wavelength tuning, along with integrating this FP-ECDL with the bidirectional temperature scanning method as part of a laser terminal, could lead to excellent applications and broader market prospects in large constellation systems, such as the U.S. “Starlink”, China’s “Star Network”, the UK’s “OneWeb”, and others.

## Figures and Tables

**Figure 1 sensors-24-01103-f001:**
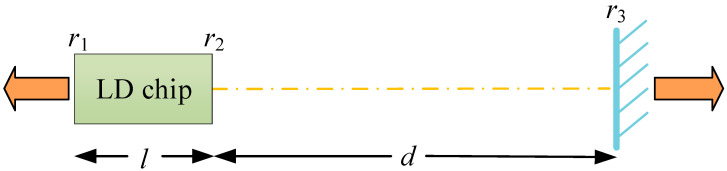
Basic model of the ECDL.

**Figure 2 sensors-24-01103-f002:**
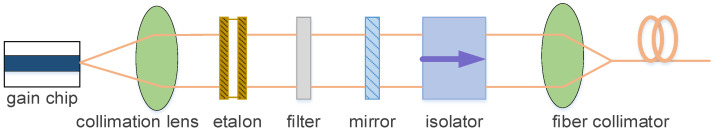
Optical path structure of the FP-ECDL.

**Figure 3 sensors-24-01103-f003:**
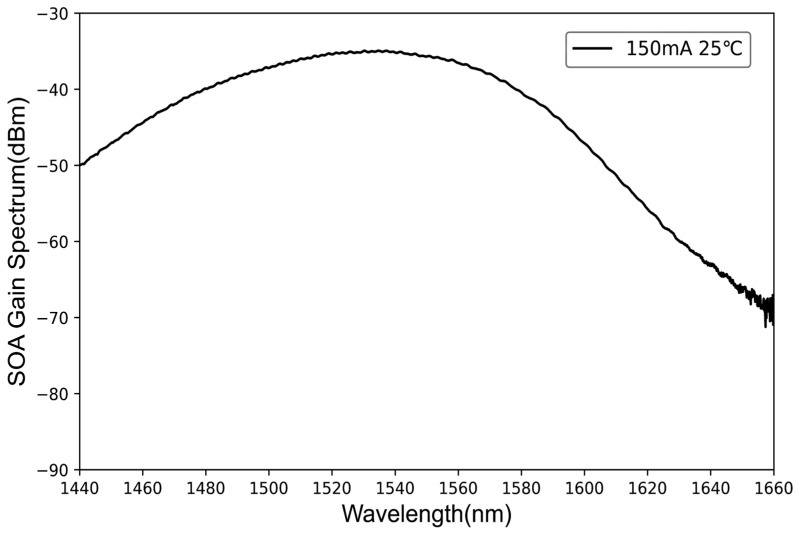
SOA gain spectrum.

**Figure 4 sensors-24-01103-f004:**
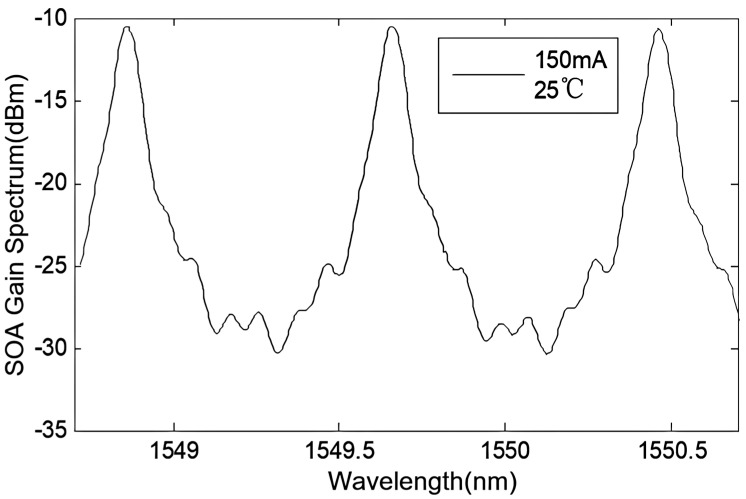
Spectrum of the gain chip after F-P etalon frequency selection (2 nm).

**Figure 5 sensors-24-01103-f005:**
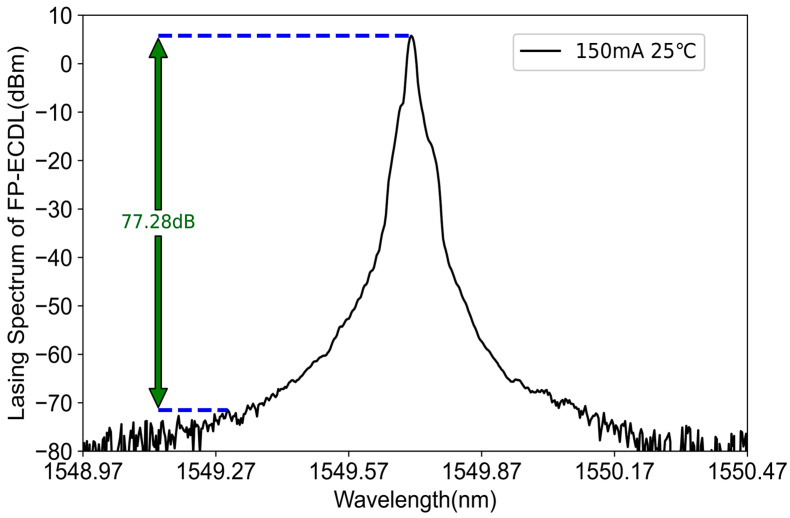
FP-ECDL laser spectrum.

**Figure 6 sensors-24-01103-f006:**
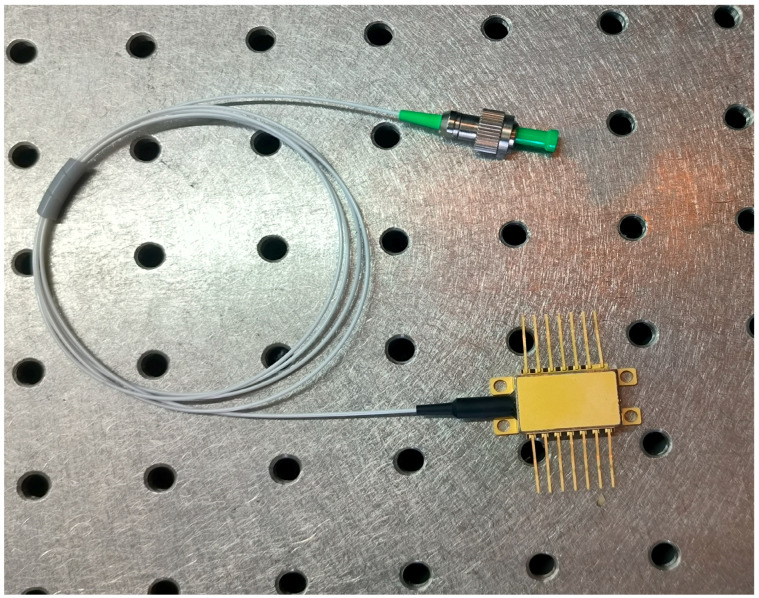
Butterfly FP-ECDL assembly.

**Figure 7 sensors-24-01103-f007:**
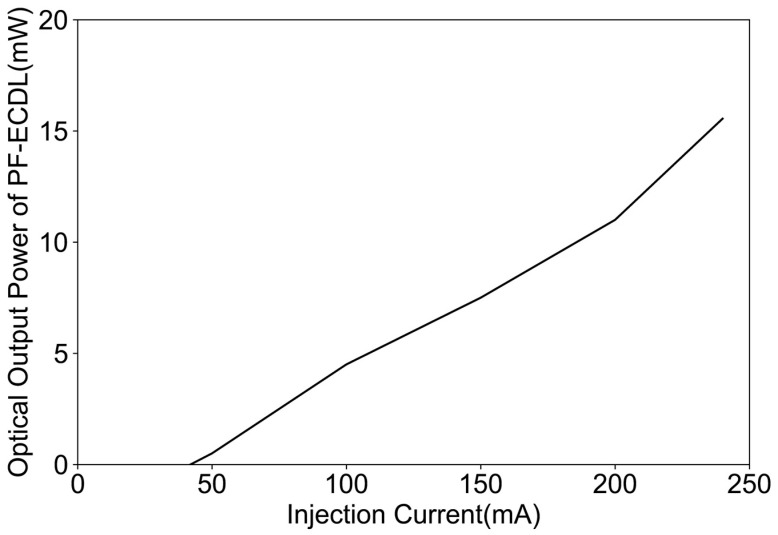
Curve for the FP-ECDL laser power variation with injection current.

**Figure 8 sensors-24-01103-f008:**
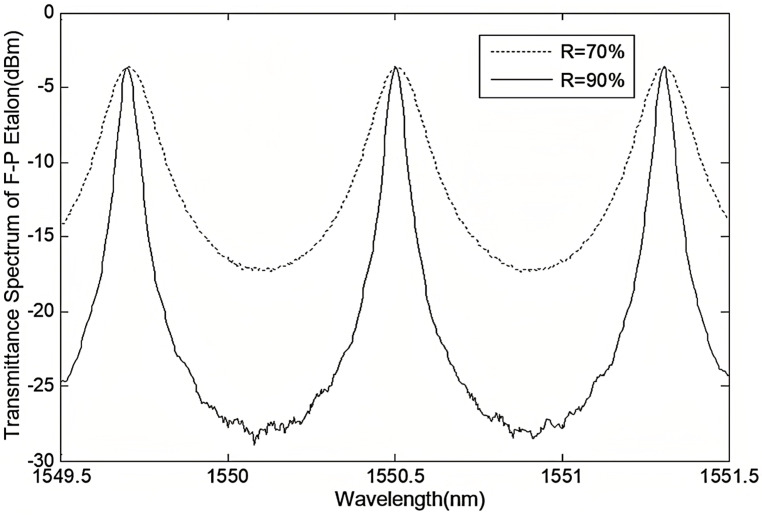
Transmission curves for two F-P etalons with different precisions.

**Figure 9 sensors-24-01103-f009:**
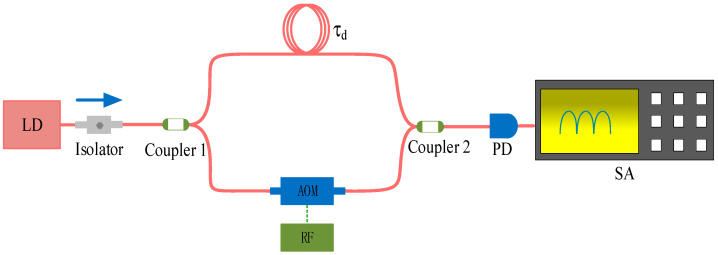
Delay self-heterodyne linewidth measurement system.

**Figure 10 sensors-24-01103-f010:**
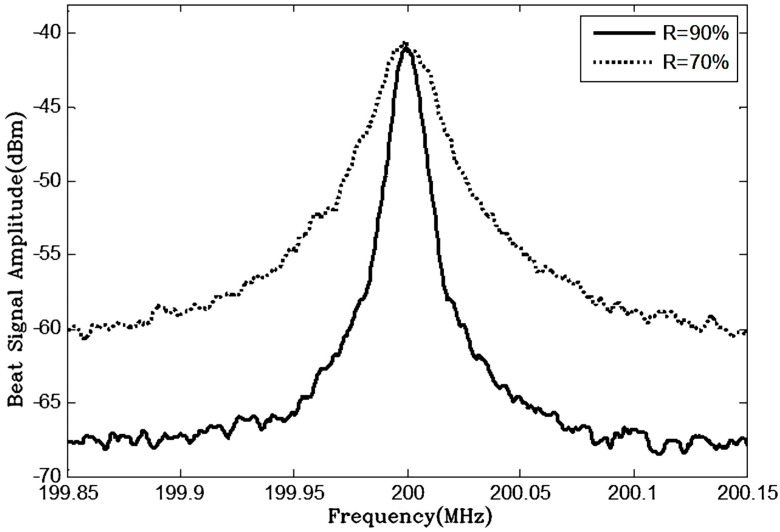
Delay self-heterodyne (30 km OPD) beat frequency testing curves of two F-P etalon precision FP-ECDLs.

**Figure 11 sensors-24-01103-f011:**
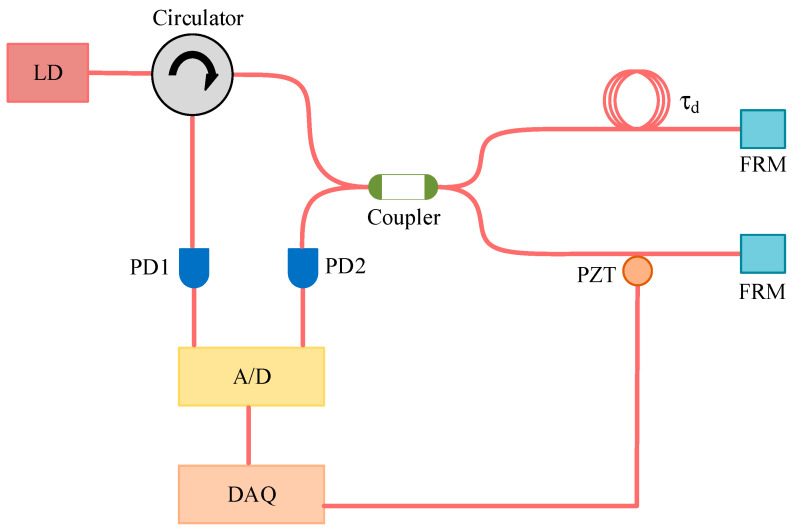
Basic structure of the phase noise measurement system.

**Figure 12 sensors-24-01103-f012:**
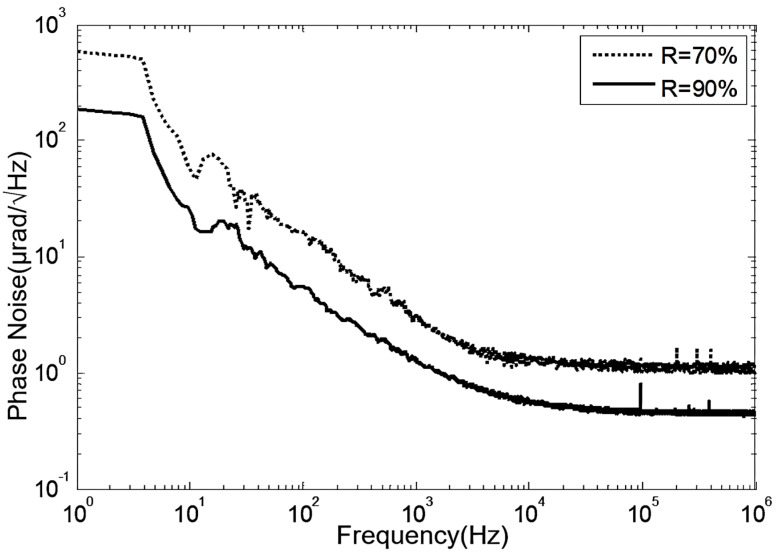
Phase noise test curves for two FP-ECDLs with different F-P etalon precisions.

**Figure 13 sensors-24-01103-f013:**
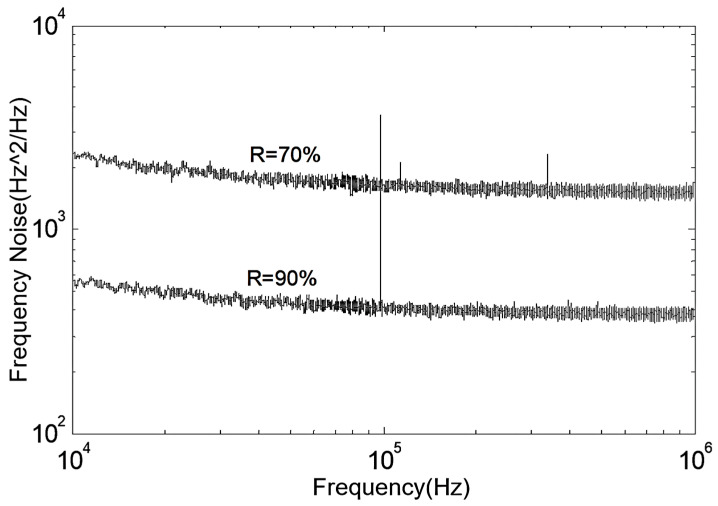
Frequency noise curves of two FP-ECDLs with different F-P etalon precisions.

**Figure 14 sensors-24-01103-f014:**
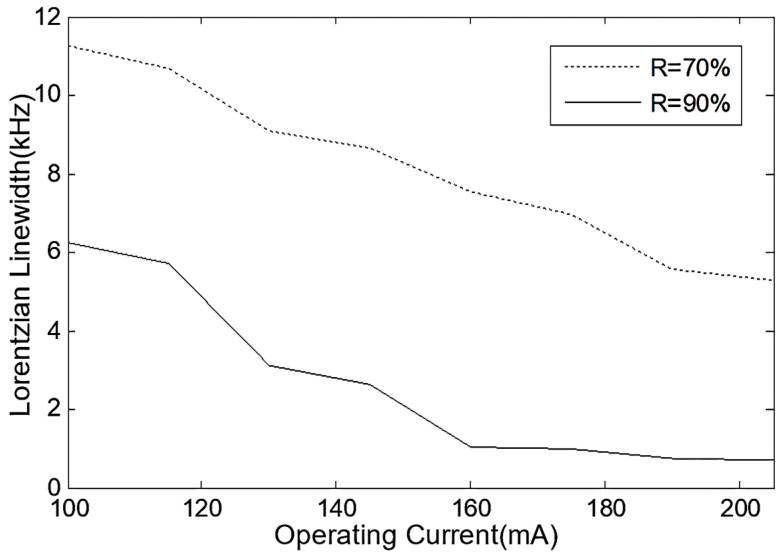
The Lorentzian linewidths of two FP-ECDLs with different F-P etalon precisions.

**Figure 15 sensors-24-01103-f015:**
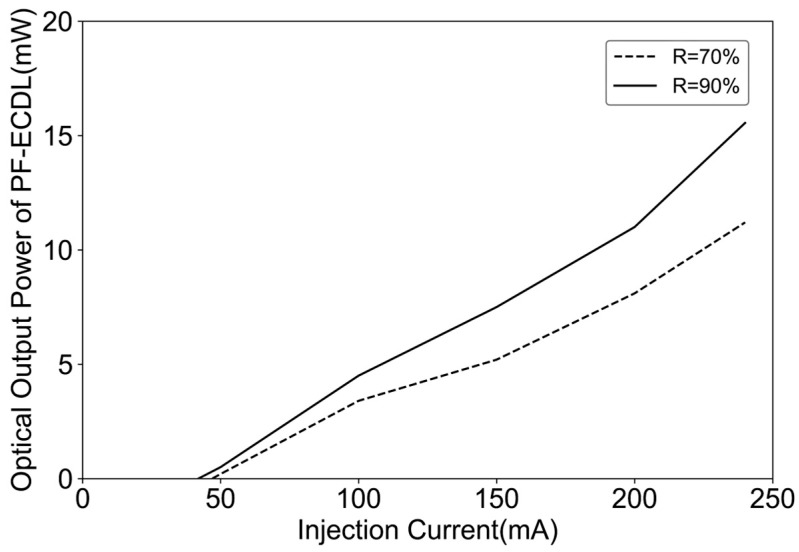
Power-injection characteristics of two FP-ECDLs with different F-P etalon precisions.

**Figure 16 sensors-24-01103-f016:**
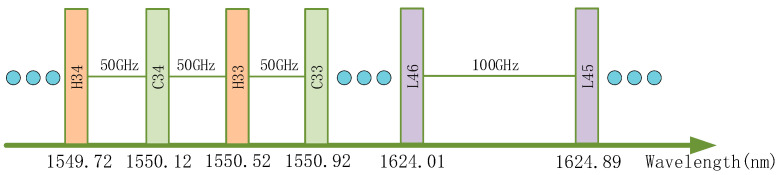
Partial channels with a 100 GHz interval.

**Figure 17 sensors-24-01103-f017:**
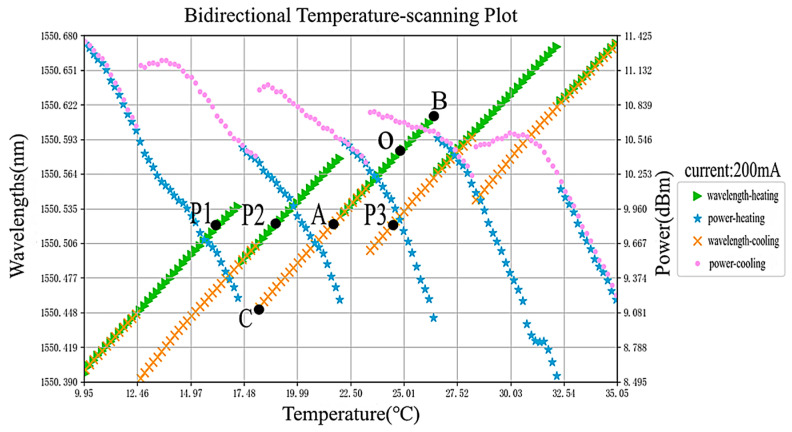
Curves of the bidirectional temperature-scanning of the FP-ECDL.

## Data Availability

Data are contained within the article.

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
