# Peer review of "Study on Linewidth and Phase Noise Characteristics of a Narrow Linewidth External Cavity Diode Laser"

_sensors, 2024, doi:10.3390/s24041103_

Round 1
Reviewer 1 Report
Comments and Suggestions for Authors
In this manuscript, the authors offer an experimental study on linewidth and phase noise characteristics of a narrow linewidth external cavity diode laser. The article aims to identify optimal conditions for narrow linewidth lasers with minimal phase and frequency noise. Furthermore, the paper introduces an innovative bidirectional temperature scanning laser method to achieve optimal laser operating point selection and mode-hop-free tuning. However, before recommending it for publication, the authors should address the following questions:
1. Consider expanding the introduction to include a brief primer on foundational concepts, aiding readers not deeply versed in the intricacies of the topic.
2. As an article focused on the experiment, detailed parameters should be provided laser diode parameters, threshold current, and optical feedback strength.
3. The influence of the external optical feedback strength on the phase noise and frequency noise was not considered in the paper.
4. In Figs. 3-10 and 3-11, the authors investigated the phase and frequency noise with two different etalon precisions at low-frequency regimes. Why did not extend their investigation to a high-frequency regime? Further, it would be nice to explore the effect of various delays and optical feedback strengths on the noise of the laser diode.
5. In Fig. 3-11, the x-axis starts from 104. Why this particular value of frequency is considered? How about the behavior at other lower and higher frequencies?
6. In most of the results the related physical explanation is not provided. I suggest adding some physical insights for various dynamics as well as changes in the phase and frequency noise.
7. It has been shown that the results significantly differ for short and long external cavity lengths, and only one specific case is presented here. How about behaviors at other short and intermediate cavity lengths? It would be nice to present some generalized studies based on a reasonable range of external cavity lengths. It would make this study more complete, and useful for applications of laser diode.
8. Dedicate a section towards the end of the paper discussing in-depth the real-world applications, challenges, and future prospects of the research findings. This would provide readers with a clearer understanding of the broader implications and relevance of the study.
Reviewer 2 Report
Comments and Suggestions for Authors
The laser design introduces an etalon and auxiliary filter for single longitudinal mode selection. Linewidth and noise characteristics are measured using delayed self-heterodyne and noise analyzer methods, yielding beat spectra and noise power spectral densities. Experimental results show a side-mode suppression ratio of 70dB and >10mW output power. Improving the etalon precision further suppresses low-frequency phase noise, reducing the linewidth. A bidirectional temperature scanning method is presented for selecting the optimal operating point and enabling mode-hop-free tuning. Overall, the laser demonstrates narrow linewidth, low noise performance, and tunability achieved via the etalon- and filter-based external cavity design. But some aspects could be further investigated and improved:
The introduction does not provide an answer to the question of why the wavelength of 1550 nm was chosen for inter-satellite communication systems.
On Line 32, it is preferable to replace the word "reception" with "receiving."
The introduction does not mention the laser's spectral linewidth requirement for inter-satellite laser communication systems. In reference [5], for example, a laser with a linewidth of 20 Hz is mentioned, while in this article, the authors discuss a linewidth of 5 kHz.
The resolution of the figures should be increased.
The relative intensity noise (RIN) performance of the laser should be illustrated as it is an important parameter for laser characterization. Without this information, the characterization of the laser system is incomplete.
Laser noise is measured in dB/Hz, while phase noise is measured in [dBc/Hz] or [(rad^2)*Hz]. However, in Figure 3-10, the unit of measurement provided is [rad/sqrt(Hz)]. Please provide an explanation for this.
When referring to equation (2-1), you cite reference [20], which does not include this equation. Equation (2-2) is also missing in reference [22].
It is unclear what is meant by “while other wavelengths of light are suppressed to varying degrees” (line 122).
Please provide a description of the Semiconductor Optical Amplifier (SOA) chip's design, as it is necessary for understanding how the lateral and transverse modes are filtered (mode discrimination, mode selection).
Specify the total length of the cavity.
Provide the laser spectrum without the etalon, output power without the etalon, and with the etalon.
In Figure 3-2, are the data provided for the case when the external cavity mirror is absent? Provide the linewidth of the main broad peaks and explain the small peaks observed in the spectrum at – 23 dBm.
Figure 3-3: What is the intermode spacing in the setup with the external cavity when there is no filter or etalon? Explain the asymmetric shape of the spectrum. Add units of measurement to the vertical axis. Since there is doubt that an SMSR of 70 dB is achieved in the study, prove that there is only one mode above 70 dB. For example, provide an additional spectrum from -30 to 0 on the vertical axis with an increased wavelength resolution.
Specify the pumping type: continuous wave (CW).
The butterfly 14-pin package does not resemble a tube in its shape (line 172).
Replace "carrier concentrations" with "carrier densities."
It remains unclear what the etalon represents. Is it two gratings or is it a Fabry-Perot etalon?
Specify the resolution of the spectrometer (line 194).
Line 195: Replace "the external cavity of two etalons" with "the external cavity for two etalons" or clarify what you meant.
Line 205: Do these algorithms use different delay line lengths?
Line 222: How is the interferometer arm length changed?
Line 253: How was the calculation of the Lorentzian linewidth performed?
Line 275: Where (or where is it regulated) is the error requirement of less than 1 pm specified?
Provide a similar figure to Figure 3-5 showing the power variation for two etalon coatings. Which etalon coating is shown in Figure 3-5?
Line 281: What is the 𝐻33 channel?
What ensures the temperature stability accuracy of 0.01 degree? Which thermoresistor is used? At what distance from the laser it is installed?
Considering the stringent power and wavelength stability requirements for inter-satellite laser communication systems operating in space environments, it would be interesting to evaluate the laser's performance under simulated thermal cycling and vibration conditions. How stable are the linewidth, noise characteristics and tunability over extreme temperature ranges and mechanical stresses? Additionally, assessing the long-term reliability over durations relevant to satellite missions (>5 years) could provide useful data on the laser's suitability. Real-time performance monitoring during accelerated life testing may reveal potential degradation mechanisms. While bidirectional temperature tuning was demonstrated, integrating fast active thermo-electric tuning could allow dynamic compensation of Doppler shifts during communication links. How may the cooling/heating elements and control system be optimized for agile wavelength control at millihertz resolution levels? Your insights on plans for future work would be greatly appreciated.
Comments on the Quality of English Language
On Line 32, it is preferable to replace the word "reception" with "receiving."
It is unclear what is meant by “while other wavelengths of light are suppressed to varying degrees” (line 122).
The text is difficult to read, it would be better to improve the style.
Round 2
Reviewer 1 Report
Comments and Suggestions for Authors
No comment.
Author Response
Thank you for the review. There were no further revisions made based on your comments.
Reviewer 2 Report
Comments and Suggestions for Authors
Please find the review in the attached file.

Round 3
Reviewer 2 Report
Comments and Suggestions for Authors
Thank you for your responses.
The authors correctly point out that “It can be observed that near the so-called 'longitudinal modes,' the waveform does not exhibit an upward and downward change but rather an overall decrease." This is indeed a valid observation when the spacing between spectral lines is smaller than the spectral resolution of the optical spectrum analyzer used. It is possible that if the authors reduce the overall length of the cavity by 2-3 times, they will be able to distinguish individual lines.
The second comment from the authors, "when changing the temperature, the variation of the highest side peak is very noticeable and does not remain stable", provides a much more satisfactory response to the reviewer's question. Indeed, only instrumental error can result in the same asymmetric shape of the spectral line near its peak (a peak of about 12dB on the left and a peak of about 20dB on the right) when the external conditions change.
It is also worth noting the possibility that the Yokogawa AQ6370D optical spectrum analyzer may not be operating in the most accurate mode when measuring such narrow spectral lines with such a large dynamic range.
Therefore, I recommend that the authors clarify to the readers the measured shape of the spectral line shown in Figure 3-4.
Furthermore, I kindly request the authors to increase the resolution (dpi) of the figures containing measurement data.
